# Do Deep Convolutional Nets Really Need to be Deep and Convolutional?

**Gregor Urban[1], Krzysztof J. Geras[2], Samira Ebrahimi Kahou[3], Ozlem Aslan[4], Shengjie Wang[5], Abdelrahman Mohamed[6], Matthai Philipose[6], Matt Richardson[6], Rich Caruana[6]**
[1]UC Irvine, USA
[2]University of Edinburgh, UK
[3]Ecole Polytechnique de Montreal, CA
[4]University of Alberta, CA
[5]University of Washington, USA
[6]Microsoft Research, USA

## Abstract

Yes, they do. This paper provides the first empirical demonstration that deep convolutional models really need to be both deep and convolutional, even when trained with methods such as distillation that allow small or shallow models of high accuracy to be trained. Although previous research showed that shallow feed-forward nets sometimes can learn the complex functions previously learned by deep nets while using the same number of parameters as the deep models they mimic, in this paper we demonstrate that the same methods cannot be used to train accurate models on CIFAR-10 unless the student models contain multiple layers of convolution. Although the student models do not have to be as deep as the teacher model they mimic, the students need multiple convolutional layers to learn functions of comparable accuracy as the deep convolutional teacher.

## 1 Introduction

Cybenko (1989) proved that a network with a large enough single hidden layer of sigmoid units can approximate any decision boundary. Empirical work, however, suggests that it can be difficult to train shallow nets to be as accurate as deep nets. Dauphin and Bengio (2013) trained shallow nets on SIFT features to classify a large-scale ImageNet dataset and found that it was difficult to train large, high-accuracy, shallow nets. A study of deep convolutional nets suggests that for vision tasks deeper models are preferred under a parameter budget (e.g. Eigen et al. (2014); He et al. (2015); Simonyan and Zisserman (2014); Srivastava et al. (2015)). Similarly, Seide et al. (2011) and Geras et al. (2015) show that deeper models are more accurate than shallow models in speech acoustic modeling. More recently, Romero et al. (2015) showed that it is possible to gain increases in accuracy in models with few parameters by training deeper, thinner nets (FitNets) to mimic much wider nets. Cohen and Shashua (2016); Liang and Srikant (2016) suggest that the representational efficiency of deep networks scales exponentially with depth, but it is unclear if this applies only to pathological problems, or is encountered in practice on data sets such as TIMIT and CIFAR.

Ba and Caruana (2014), however, demonstrated that shallow nets sometimes can learn the functions learned by deep nets, even when restricted to the same number of parameters as the deep nets. They did this by first training state-of-the-art deep models, and then training shallow models to mimic the deep models. Surprisingly, and for reasons that are not well understood, the shallow models learned more accurate functions when trained to mimic the deep models than when trained on the original data used to train the deep models. In some cases shallow models trained this way were as accurate as state-of-the-art deep models. But this demonstration was made on the TIMIT speech recognition benchmark. Although their deep teacher models used a convolutional layer, convolution is less important for TIMIT than it is for other domains such as image classification.

Ba and Caruana (2014) also presented results on CIFAR-10 which showed that a shallow model could learn functions almost as accurate as deep convolutional nets. Unfortunately, the results on CIFAR-10 are less convincing than those for TIMIT. To train accurate shallow models on CIFAR-10

they had to include at least one convolutional layer in the shallow model, and increased the number of parameters in the shallow model until it was 30 times larger than the deep teacher model. Despite this, the shallow convolutional student model was several points less accurate than a teacher model that was itself several points less accurate than state-of-the-art models on CIFAR-10.

In this paper we show that the methods Ba and Caruana used to train shallow students to mimic deep teacher models on TIMIT do not work as well on problems such as CIFAR-10 where multiple layers of convolution are required to train accurate teacher models. If the student models have a similar number of parameters as the deep teacher models, high accuracy can not be achieved without multiple layers of convolution even when the student models are trained via distillation.

To ensure that the shallow student models are trained as accurately as possible, we use Bayesian optimization to thoroughly explore the space of architectures and learning hyperparameters. Although this combination of distillation and hyperparameter optimization allows us to train the most accurate shallow models ever trained on CIFAR-10, the shallow models still are not as accurate as deep models. *Our results clearly suggest that deep convolutional nets do, in fact, need to be both deep and convolutional, even when trained to mimic very accurate models via distillation (Hinton et al., 2015).*

## 2    TRAINING SHALLOW NETS TO MIMIC DEEPER CONVOLUTIONAL NETS

In this paper, we revisit the CIFAR-10 experiments in Ba and Caruana (2014). Unlike in that work, here we compare shallow models to state-of-the-art deep convolutional models, and restrict the number of parameters in the shallow student models to be comparable to the number of parameters in the deep convolutional teacher models. Because we anticipated that our results might be different, we follow their approach closely to eliminate the possibility that the results differ merely because of changes in methodology. Note that the goal of this paper is *not* to train models that are small or fast as in Bucila et al. (2006), Hinton et al. (2015), and Romero et al. (2015), but to examine if shallow models can be as accurate as deep convolutional models given the same parameter budget.

There are many steps required to train shallow student models to be as accurate as possible: train state-of-the-art deep convolutional teacher models, form an ensemble of the best deep models, collect and combine their predictions on a large transfer set, and then train carefully optimized shallow student models to mimic the teacher ensemble. For negative results to be informative, it is important that each of these steps be performed as well as possible. In this section we describe the experimental methodology in detail. Readers familiar with distillation (model compression), training deep models on CIFAR-10, data augmentation, and Bayesian hyperparameter optimization may wish to skip to the empirical results in Section 3.

### 2.1    MODEL COMPRESSION AND DISTILLATION

The key idea behind model compression is to train a compact model to approximate the function learned by another larger, more complex model. Bucila et al. (2006) showed how a single neural net of modest size could be trained to mimic a much larger ensemble. Although the *small* neural nets contained $1000\times$ fewer parameters, often they were as accurate as the large ensembles they were trained to mimic.

Model compression works by passing unlabeled data through the large, accurate teacher model to collect the real-valued scores it predicts, and then training a student model to mimic these scores. Hinton et al. (2015) generalized the methods of Bucila et al. (2006) and Ba and Caruana (2014) by incorporating a parameter to control the relative importance of the soft targets provided by the teacher model to the hard targets in the original training data, as well as a temperature parameter that regularizes learning by pushing targets towards the uniform distribution. Hinton et al. (2015) also demonstrated that much of the knowledge passed from the teacher to the student is conveyed as *dark knowledge* contained in the relative scores (probabilities) of outputs corresponding to *other* classes, as opposed to the scores given to just the output for the one correct class.

Surprisingly, distillation often allows smaller and/or shallower models to be trained that are nearly as accurate as the larger, deeper models they are trained to mimic, yet these same small models are not as accurate when trained on the 1-hot hard targets in the original training set. The reason for this is not yet well understood. Similar compression and distillation methods have also successfully

been used in speech recognition (e.g. Chan et al. (2015); Geras et al. (2015); Li et al. (2014)) and reinforcement learning Parisotto et al. (2016); Rusu et al. (2016). Romero et al. (2015) showed that distillation methods can be used to train small students that are more accurate than the teacher models by making the student models deeper, but thinner, than the teacher model.

## 2.2 MIMIC LEARNING VIA L2 REGRESSION ON LOGITS

We train shallow mimic nets using data labeled by an ensemble of deep teacher nets trained on the original 1-hot CIFAR-10 training data. The deep teacher models are trained in the usual way using softmax outputs and cross-entropy cost function. Following Ba and Caruana (2014), the student mimic models are not trained with cross-entropy on the ten $p$ values where $p_k = e^{z_k} / \sum_j e^{z_j}$ output by the softmax layer from the deep teacher model, but instead are trained on the un-normalized log probability values $z$ (the logits) *before* the softmax activation. Training on the logarithms of predicted probabilities (logits) helps provide the *dark knowledge* that regularizes students by placing emphasis on the relationships learned by the teacher model across all of the outputs.

As in Ba and Caruana (2014), the student is trained as a regression problem given training data $\{(x^{(1)}, z^{(1)}),...,(x^{(T)}, z^{(T)})\}$:

$$\mathcal{L}(W) = \frac{1}{T} \sum_t ||g(x^{(t)}; W) - z^{(t)}||_2^2, \qquad (1)$$

where $W$ represents all of the weights in the network, and $g(x^{(t)}; W)$ is the model prediction on the $t^{th}$ training data sample.

## 2.3 USING A LINEAR BOTTLENECK TO SPEED UP TRAINING

A shallow net has to have more hidden units in each layer to match the number of parameters in a deep net. Ba and Caruana (2014) found that training these wide, shallow mimic models with backpropagation was slow, and introduced a linear bottleneck layer between the input and non-linear layers to speed learning. The bottleneck layer speeds learning by reducing the number of parameters that must be learned, but does not make the model deeper because the linear terms can be absorbed back into the non-linear weight matrix after learning. See Ba and Caruana (2014) for details. To match their experiments we use linear bottlenecks when training student models with 0 or 1 convolutional layers, but did not find the linear bottlenecks necessary when training student models with more than 1 convolutional layer.

## 2.4 BAYESIAN HYPERPARAMETER OPTIMIZATION

The goal of this work is to determine empirically if shallow nets can be trained to be as accurate as deep convolutional models using a similar number of parameters in the deep and shallow models. If we succeed in training a shallow model to be as accurate as a deep convolutional model, this provides an existence proof that shallow models can represent and learn the complex functions learned by deep convolutional models. If, however, we are unable to train shallow models to be as accurate as deep convolutional nets, we might fail only because we did not train the shallow nets well enough.

In all our experiments we employ Bayesian hyperparameter optimization using Gaussian process regression to ensure that we thoroughly and objectively explore the hyperparameters that govern learning. The implementation we use is Spearmint (Snoek et al., 2012). The hyperparameters we optimize with Bayesian optimization include the initial learning rate, momentum, scaling of the initial random weights, scaling of the inputs, and terms that determine the width of each of the network's layers (i.e. number of convolutional filters and neurons). More details of the hyperparameter optimization can be found in Sections 2.5, 2.7, 2.8 and in the Appendix.

## 2.5 TRAINING DATA AND DATA AUGMENTATION

The CIFAR-10 (Krizhevsky, 2009) data set consists of a set of natural images from 10 different object classes: airplane, automobile, bird, cat, deer, dog, frog, horse, ship, truck. The dataset is a labeled subset of the 80 million tiny images dataset (Torralba et al., 2008) and is divided into 50,000 train and

10,000 test images. Each image is 32×32 pixels in 3 color channels, yielding input vectors with 3072 dimensions. We prepared the data by subtracting the mean and dividing by the standard deviation of each image vector. We train all models on a subset of 40,000 images and use the remaining 10,000 images as the validation set for the Bayesian optimization. The final trained models only used 80% of the theoretically available training data (as opposed to retraining on all of the data after hyperparameter optimization).

We employ the HSV-data augmentation technique as described by Snoek et al. (2015). Thus we shift hue, saturation and value by uniform random values: $\Delta_h \sim U(-D_h, D_h)$, $\Delta_s \sim U(-D_s, D_s)$, $\Delta_v \sim U(-D_v, D_v)$. Saturation and value values are scaled globally: $a_s \sim U(\frac{1}{1+A_s}, 1 + A_s), a_v \sim U(\frac{1}{1+A_v}, 1 + A_v)$. The five constants $D_h, D_s, D_v, A_s, A_v$ are treated as additional hyperparameters in the Bayesian hyperparameter optimization.

All training images are mirrored left-right randomly with a probability of $0.5$. The input images are further scaled and jittered randomly by cropping windows of size 24×24 up to 32×32 at random locations and then scaling them back to 32×32. The procedure is as follows: we sample an integer value $S \sim U(24, 32)$ and then a pair of integers $x, y \sim U(0, 32 - S)$. The transformed resulting image is $R = f_{\text{spline},3}(I[x : x + S, y : y + S])$ with $I$ denoting the original image and $f_{\text{spline},3}$ denoting the 3rd order spline interpolation function that maps the 2D array back to 32×32 (applied to the three color channels separately).

All data augmentations for the teacher models are computed on the fly using different random seeds. For student models trained to mimic the ensemble (see Section 2.7 for details of the ensemble teacher model), we pre-generated 160 epochs worth of randomly augmented training data, evaluated the ensemble's predictions (logits) on these samples, and saved all data and predictions to disk. All student models thus see the same training data in the same order. The parameters for HSV-augmentation in this case had to be selected beforehand; we chose to use the settings found with the best single model ($D_h = 0.06, D_s = 0.26, D_v = 0.20, A_s = 0.21, A_v = 0.13$). Pre-saving the logits and augmented data is important to reduce the computational cost at training time, and to ensure that all student models see the same training data

Because augmentation allows us to generate large training sets from the original 50,000 images, we use augmented data as the transfer set for model compression. No extra unlabeled data is required.

## 2.6 LEARNING-RATE SCHEDULE

We train all models using SGD with Nesterov momentum. The initial learning rate and momentum are chosen by Bayesian optimization. The learning rate is reduced according to the evolution of the model's validation error: it is halved if the validation error does not drop for ten epochs in a row. It is not reduced within the next eight epochs following a reduction step. Training ends if the error did not drop for 30 epochs in a row or if the learning rate was reduced by a factor of more than 2000 in total.

This schedule provides a way to train the highly varying models in a fair manner (it is not feasible to optimize all of the parameters that define the learning schedule). It also decreases the time spent to train each model compared to using a hand-selected overestimate of the number of epochs to train, thus allowing us to train more models in the hyperparameter search.

## 2.7 SUPER TEACHER: AN ENSEMBLE OF 16 DEEP CONVOLUTIONAL CIFAR-10 MODELS

One limitation of the CIFAR-10 experiments performed in Ba and Caruana (2014) is that the teacher models were not state-of-the-art. The best deep models they trained on CIFAR-10 had only 88% accuracy, and the ensemble of deep models they used as a teacher had only 89% accuracy. The accuracies were not state-of-the-art because they did not use augmentation and because their deepest models had only three convolutional layers. Because our goal is to determine if shallow models can be as accurate as deep convolutional models, it is important that the deep models we compare to (and use as teachers) are as accurate as possible.

We train deep neural networks with eight convolutional layers, three intermittent max-pooling layers and two fully-connected hidden layers. We include the size of these layers in the hyperparameter optimization, by allowing the first two convolutional layers to contain from 32 to 96 filters each, the next two layers to contain from 64 to 192 filters, and the last four convolutional layers to contain

from 128 to 384 filters. The two fully-connected hidden layers can contain from 512 to 1536 neurons. We parametrize these model-sizes by four scalars (the layers are grouped as 2-2-4) and include the scalars in the hyperparameter optimization. All models are trained using Theano (Bastien et al., 2012; Bergstra et al., 2010).

We optimize eighteen hyperparameters overall: initial learning rate on $[0.01, 0.05]$, momentum on $[0.80, 0.91]$, $l_2$ weight decay on $[5 \cdot 10^{-5}, 4 \cdot 10^{-4}]$, initialization coefficient on $[0.8, 1.35]$ which scales the initial weights of the CNN, four separate dropout rates, five constants controlling the HSV data augmentation, and the four scaling constants controlling the networks' layer widths. The learning rate and momentum are optimized on a log-scale (as opposed to linear scale) by optimizing the exponent with appropriate bounds, e.g. $LR = e^{-x}$ optimized over $x$ on $[3.0, 4.6]$. See the Appendix for more details about hyperparameter optimization.

We trained 129 deep CNN models with Spearmint. The best model obtained an accuracy of 92.78%; the fifth best achieved 92.67%. See Table 1 for the sizes and architectures of the three best models.

We are able to construct a more accurate model on CIFAR-10 by forming an ensemble of multiple deep convolutional neural nets, each trained with different hyperparameters, and each seeing slightly different training data (as the augmentation parameters vary). We experimented with a number of ensembles of the many deep convnets we trained, using accuracy on the validation set to select the best combination. The final ensemble contained 16 deep convnets and had an accuracy of 94.0% on the validation set, and 93.8% on the final test set. We believe this is among the top published results for deep learning on CIFAR-10. The ensemble averages the logits predicted by each model before the softmax layers.

We used this very accurate ensemble model as the teacher model to label the data used to train the shallower student nets. As described in Section 2.2, the logits (the scores just prior to the final softmax layer) from each of the CNN teachers in the ensemble model are averaged for each class, and the average logits are used as final regression targets to train the shallower student neural nets.

## 2.8    Training Shallow Student Models to Mimic an Ensemble of Deep Convolutional Models

We trained student mimic nets with 1, 3.16[1], 10 and 31.6 million trainable parameters on the pre-computed augmented training data (Section 2.5) that was re-labeled by the teacher ensemble (Section 2.7). For each of the four student sizes we trained shallow fully-connected student MLPs containing 1, 2, 3, 4, or 5 layers of non-linear units (ReLU), and student CNNs with 1, 2, 3 or 4 convolutional layers. The convolutional student models also contain one fully-connected ReLU layer. Models with zero or only one convolutional layer contain an additional linear bottleneck layer to speed up learning (cf. Section 2.3). We did not need to use a bottleneck to speed up learning for the deeper models as the number of learnable parameters is naturally reduced by the max-pooling layers.

The student CNNs use max-pooling and Bayesian optimization controls the number of convolutional filters and hidden units in each layer. The hyperparameters we optimized in the student models are: initial learning rate, momentum, scaling of the initially randomly distributed learnable parameters, scaling of all pixel values of the input, and the scale factors that control the width of all hidden and convolutional layers in the model. Weights are initialized as in Glorot and Bengio (2010). We intentionally do not optimize and do not make use of weight decay and dropout when training student models because preliminary experiments showed that these consistently reduced the accuracy of student models by several percent. Please refer to the Appendix for more details on the individual architectures and hyperparameter ranges.

## 3    Empirical Results

Table 1 summarizes results after Bayesian hyperparameter optimization for models trained on the original 0/1 hard CIFAR-10 labels. All of these models use weight decay and are trained with the dropout hyperparameters included in the Bayesian optimization. The table shows the accuracy of the best three deep convolutional models we could train on CIFAR-10, as well as the accuracy of

---

[1]3.16 ≈ Sqrt(10) falls halfway between 1 and 10 on log scale.

Table 1: Accuracy on CIFAR-10 of shallow and deep models trained on the original 0/1 hard class labels using Bayesian optimization with dropout and weight decay. Key: c = convolution layer; mp = max-pooling layer; fc = fully-connected layer; lfc = linear bottleneck layer; exponents indicate repetitions of a layer. The last two models (*) are numbers reported by Ba and Caruana (2014). The models with 1-4 convolutional layers at the top of the table are included for comparison with student models of similar architecture in Table 2 . All of the *student* models in Table 2 with 1, 2, 3, and 4 convolutional layers are more accurate than their counterparts in this table that are trained on the original 0/1 hard targets — as expected distillation yields shallow models of higher accuracy than shallow models trained on the original training data.

| Model | Architecture | # parameters | Accuracy |
|---|---|---|---|
| 1 conv. layer | c-mp-lfc-fc | 10M | 84.6% |
| 2 conv. layer | c-mp-c-mp-fc | 10M | 88.9% |
| 3 conv. layer | c-mp-c-mp-c-mp-fc | 10M | 91.2% |
| 4 conv. layer | c-mp-c-c-mp-c-mp-fc | 10M | 91.75% |
| Teacher CNN $1^{st}$ | $76c^2$-mp-$126c^2$-mp-$148c^4$-mp-$1200fc^2$ | 5.3M | 92.78% |
| Teacher CNN $2^{nd}$ | $96c^2$-mp-$171c^2$-mp-$128c^4$-mp-$512fc^2$ | 2.5M | 92.77% |
| Teacher CNN $3^{rd}$ | $54c^2$-mp-$158c^2$-mp-$189c^4$-mp-$1044fc^2$ | 5.8M | 92.67% |
| Ensemble of 16 CNNs | $c^2$-mp-$c^2$-mp-$c^4$-mp-$fc^2$ | 83.4M | 93.8% |
| Teacher CNN (*) | 128c-mp-128c-mp-128c-mp-1k fc | 2.1M | 88.0% |
| Ensemble, 4 CNNs (*) | 128c-mp-128c-mp-128c-mp-1k fc | 8.6M | 89.0% |

Table 2: Comparison of student models with varying number of convolutional layers trained to mimic the ensemble of 16 deep convolutional CIFAR-10 models in Table 1 . The best performing student models have $3-4$ convolutional layers and $10M-31.6M$ parameters. The student models in this table are more accurate than the models of the same architecture in Table 1 that were trained on the original 0/1 hard targets — shallow models trained with distillation are more accurate than shallow models trained on 0/1 hard targets. The student model trained by Ba and Caruana (2014) is shown in the last line for comparison; it is less accurate and much larger than the student models trained here that also have 1 convolutional layer.

| | 1 M | 3.16 M | 10 M | 31.6 M | 70 M |
|---|---|---|---|---|---|
| Bottleneck, 1 hidden layer | 65.8% | 68.2% | 69.5% | 70.2% | – |
| 2 hidden layers | 66.2% | 70.9% | 73.4% | 74.3% | – |
| 3 hidden layers | 66.8% | 71.0% | 73.0% | 73.9% | – |
| 4 hidden layers | 66.7% | 69.5% | 71.6% | 72.0% | – |
| 5 hidden layers | 66.4% | 70.0% | 71.4% | 71.5% | – |
| 1 conv. layer, 1 max-pool, Bottleneck | 84.5% | 86.3% | 87.3% | 87.7% | – |
| 2 conv. layers, 2 max-pool | 87.9% | 89.3% | 90.0% | 90.3% | – |
| 3 conv. layers, 3 max-pool | 90.7% | 91.6% | 91.9% | 92.3% | – |
| 4 conv. layers, 3 max-pool | 91.3% | 91.8% | 92.6% | 92.6% | – |
| SNN-ECNN-MIMIC-30k 128c-p-1200L-30k trained on ensemble (Ba and Caruana, 2014) | – | – | – | – | 85.8% |

the ensemble of 16 deep CNNs. For comparison, the accuracy of the ensemble trained by Ba and Caruana (2014)) is included at the bottom of the table.

Table 2 summarizes the results after Bayesian hyperparameter optimization for **student models** of different depths and number of parameters trained on **soft targets** (average logits) to mimic the teacher ensemble of 16 deep CNNs. For comparison, the student model trained by Ba and Caruana (2014) also is shown.

The first four rows in Table 1 show the accuracy of convolutional models with 10 million parameters and 1, 2, 3, and 4 convolutional layers. The accuracies of these same architectures with 1M, 3.16M, 10M, and 31.6M parameters when trained as students on the soft targets predicted by the teacher ensemble are shown in Table 2. Comparing the accuracies of the models with 10 million parameters in both tables, we see that training student models to mimic the ensemble leads to significantly better accuracy in every case. The gains are more pronounced for shallower models, most likely because their learnable internal representations do not naturally lead to good generalization in this task when trained on the 0/1 hard targets: the difference in accuracy for models with one convolutional layer is 2.7% (87.3% vs. 84.6%) and only 0.8% (92.6% vs. 91.8%) for models with four convolutional layers.

Figure 1 summarizes the results in Table 2 for student models of different depth, number of convolutional layers, and number of parameters when trained to mimic the ensemble teacher model. Student models trained on the ensemble logits are able to achieve accuracies previously unseen on CIFAR-10 for models with so few layers. Also, it is clear that there is a huge gap between the convolutional student models at the top of the figure, and the non-convolutional student models at the bottom of the figure: the most accurate student MLP has accuracy less than 75%, while the least accurate convolutional student model with the same number of parameters but only one convolutional layer has accuracy above 87%. And the accuracy of the convolutional student models increases further as more layers of convolution are added. Interestingly, the most accurate student MLPs with no convolutional layers have only 2 or 3 hidden layers; the student MLPs with 4 or 5 hidden layers are not as accurate.

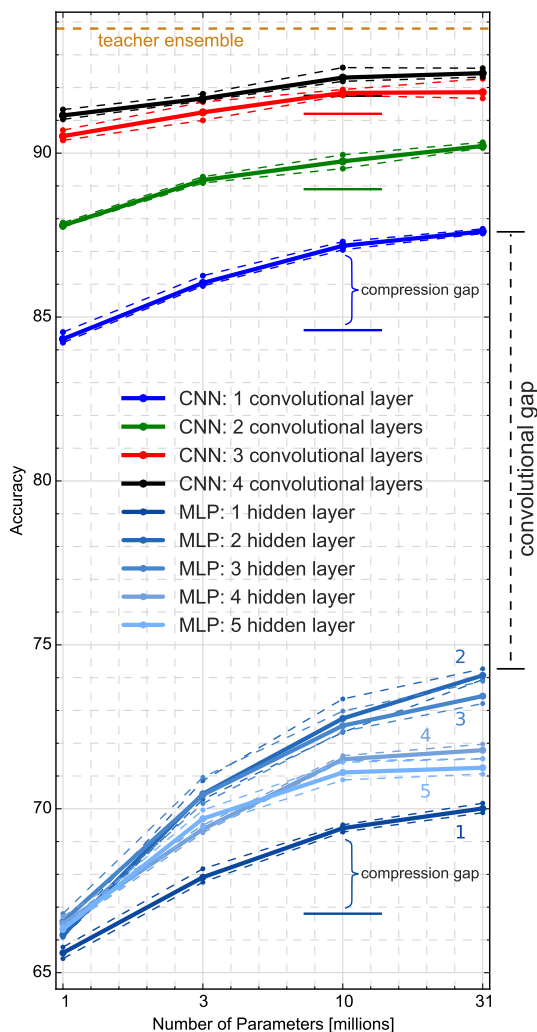

Figure 1: Accuracy of student models with different architectures trained to mimic the CIFAR10 ensemble. The average performance of the five best models of each hyperparameter-optimization experiment is shown, together with dashed lines indicating the accuracy of the best and the fifth best model from each setting. The short horizontal lines at 10M parameters are the accuracy of models trained without compression on the original 0/1 hard targets.

Comparing the student MLP with only one hidden layer (bottom of the graph) to the student CNN with 1 convolutional layer clearly suggests that convolution is critical for this problem even when models are trained via distillation, and that it is very unlikely that a shallow non-convolutional model with 100 million parameters or less could ever achieve accuracy comparable to a convolutional model. It appears that if convolution is critical for teacher models trained on the original 0/1 hard targets, it

is likely to be critical for student models trained to mimic these teacher models. Adding depth to the student MLPs without adding convolution does not significantly close this "convolutional gap".

Furthermore, comparing student CNNs with 1, 2, 3, and 4 convolutional layers, it is clear that CNN students benefit from multiple convolutional layers. Although the students do not need as many layers as teacher models trained on the original 0/1 hard targets, accuracy increases significantly as multiple convolutional layers are added to the model. For example, the best student with only one convolutional layer has 87.7% accuracy, while the student with the same number of parameters (31M) and 4 convolutional layers has 92.6% accuracy.

Figure 1 includes short horizontal lines at 10M parameters indicating the accuracy of non-student models trained on the original 0/1 hard targets instead of on the soft targets. This "compression gap" is largest for shallower models, and as expected disappears as the student models become architecturally more similar to the teacher models with multiple layers of convolution. The benefits of distillation are most significant for shallow models, yielding an increase in accuracy of 3% or more.

One pattern that is clear in the graph is that all student models benefit when the number of parameters increases from 1 million to 31 million parameters. It is interesting to note, however, that the largest student (31M) with a one convolutional layer is less accurate than the smallest student (1M) with two convolutional layers, further demonstrating the value of depth in convolutional models.

In summary, depth-constrained student models trained to mimic a high-accuracy ensemble of deep convolutional models perform better than similar models trained on the original hard targets (the "compression" gaps in Figure 1), student models need at least 3-4 convolutional layers to have high accuracy on CIFAR-10, shallow students with no convolutional layers perform poorly on CIFAR-10, and student models need at least 3-10M parameters to perform well. We are not able to compress deep convolutional models to *shallow* student models without significant loss of accuracy.

We are currently running a reduced set of experiments on ImageNet, though the chances of shallow models performing well on a more challenging problem such as ImageNet appear to be slim.

## 4 DISCUSSION

Although we are not able to train shallow models to be as accurate as deep models, the models trained via distillation are the most accurate models of their architecture ever trained on CIFAR-10. For example, the best single-layer fully-connected MLP (no convolution) we trained achieved an accuracy of 70.2%. We believe this to be the most accurate shallow MLP ever reported for CIFAR-10 (in comparison to 63.1% achieved by Le et al. (2013), 63.9% by Memisevic et al. (2015) and 64.3% by Geras and Sutton (2015)). Although this model cannot compete with convolutional models, clearly distillation helps when training models that are limited by architecture and/or number of parameters. Similarly, the student models we trained with 1, 2, 3, and 4 convolutional layers are, we believe, the most accurate convnets of those depths reported in the literature. For example, the ensemble teacher model in Ba and Caruana (2014) was an ensemble of four CNNs, each of which had 3 convolutional layers, but only achieved 89% accuracy, whereas the single student CNNs we train via distillation achieve accuracies above 90% with only 2 convolutional layers, and above 92% with 3 convolutional layers. The only other work we are aware of that achieves comparable high accuracy with non-convolutional MLPs is recent work by Lin et al. (2016). They train multi-layer Z-Lin networks, and use a powerful form of data augmentation based on deformations that we did not use.

Interestingly, we noticed that mimic networks perform consistently worse when trained using dropout. This surprised us, and suggests that training student models on the soft-targets from a teacher provides significant regularization for the student models obviating the need for extra regularization methods such as dropout. This is consistent with the observation made by Ba and Caruana (2014) that student mimic models did not seem to overfit. Hinton et al. (2015) claim that soft targets convey more information per sample than Boolean hard targets. The also suggest that the dark knowledge in the soft targets for other classes further helped regularization, and that early stopping was unnecessary. Romero et al. (2015) extend distillation by using the intermediate representations learned by the teacher as hints to guide training deep students, and teacher confidences further help regularization by providing a measure of sample simplicity to the student, akin to curriculum learning. In other work, Pereyra et al. (2017) suggest that the soft targets provided by a teacher provide a form of confidence penalty that penalizes low entropy distributions and label smoothing, both of which improve regularization by maintaining a reasonable ratio between the logits of incorrect classes.

Zhang et al. (2016) question the traditional view of regularization in deep models. Although they do not discuss distillation, they suggest that in deep learning traditional function approximation appears to be deeply intertwined with massive memorization. The multiple soft targets used to train student models have a high information density (Hinton et al., 2015) and thus provide regularization by reducing the impact of brute-force memorization.

## 5 CONCLUSIONS

We train shallow nets with and without convolution to mimic state-of-the-art deep convolutional nets. If one controls for the number of learnable parameters, nets containing a single fully-connected non-linear layer and no convolutional layers are not able to learn functions as accurate as deeper convolutional models. This result is consistent with those reported in Ba and Caruana (2014). However, we also find that shallow nets that contain only 1-2 convolutional layers also are unable to achieve accuracy comparable to deeper models *if the same number of parameters are used in the shallow and deep models*. Deep convolutional nets are significantly more accurate than shallow convolutional models, given the same parameter budget. We do, however, see evidence that model compression allows accurate models to be trained that are shallower and have fewer convolutional layers than the deep convolutional architectures needed to learn high-accuracy models from the original 1-hot hard-target training data. The question remains why extra layers are required to train accurate models from the original training data.

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

## 6 APPENDIX

### 6.1 DETAILS OF TRAINING THE TEACHER MODELS

Weights of trained nets are initialized as in Glorot and Bengio (2010). The models trained in Section 2.7 contain eight convolutional layers organized into three groups (2-2-4) and two fully-connected hidden layers. The Bayesian hyperparameter optimization controls four constants $C_1, C_2, C_3, H_1$ all in the range $[0, 1]$ that are then linearly transformed to the number of filters/neurons in each layer. The hyperparameters for which ranges were not shown in Section 2.7 are: the four separate dropout rates ($DOc_1, DOc_2, DOc_3, DOf$) and the five constants $D_h, D_s, D_v, A_s, A_v$ controlling the HSV data augmentation. The ranges we selected are $DOc_1 \in [0.1, 0.3], DOc_2 \in [0.25, 0.35], DOc_3 \in [0.3, 0.44], DOf_1 \in [0.2, 0.65], DOf_2 \in [0.2, 0.65], D_h \in [0.03, 0.11], D_s \in [0.2, 0.3], D_v \in [0.0, 0.2], A_s \in [0.2, 0.3], A_v \in [0.03, 0.2]$, partly guided by Snoek et al. (2015) and visual inspection of the resulting augmentations.

The number of filters and hidden units for the models have the following bounds:
1 conv. layer: 50 - 500 filters, 200 - 2000 hidden units, number of units in bottleneck is the dependent variable.
2 conv. layers: 50 - 500 filters, 100 - 400 filters, number of hidden units is the dependent variable.
3 conv. layers: 50 - 500 filters (layer 1), 100 - 300 filters (layers 2-3), # of hidden units is dependent the variable.
4 conv. layers: 50 - 300 filters (layers 1-2), 100 - 300 filters (layers 3-4), # of hidden units is the dependent variable.

All convolutional filters in the model are sized $3 \times 3$, max-pooling is applied over windows of $2 \times 2$ and we use ReLU units throughout all our models. We apply dropout after each max-pooling layer with the three rates $DOc_1, DOc_2, DOc_3$ and after each of the two fully-connected layers with the same rate $DOf$.

Table 3: Optimization bounds for student models. (Models trained on 0/1 hard targets were described in Sections 6.1 and 6.2.) Abbreviations: **fc** (fully-connected layer, ReLu), **c** (convolutional, ReLu), **linear** (fully-connected bottleneck layer, linear activation function), **dependent** (dependent variable, chosen s.t. parameter budget is met).

| | $1^{st}$ layer | $2^{nd}$ layer | $3^{rd}$ layer | $4^{th}$ layer | $5^{th}$ layer |
|---|---|---|---|---|---|
| No conv. layer (1M) | 500 - 5000 (fc) | dependent (linear) | | | |
| No conv. layer (3.1M) | 1000 - 20000 (fc) | dependent (linear) | | | |
| No conv. layer (10M) | 5000 - 30000 (fc) | dependent (linear) | | | |
| No conv. layer (31M) | 5000 - 45000 (fc) | dependent (linear) | | | |
| 1 conv. layer (1M) | 40 - 150 (c) | dependent (linear) | 200 - 1600 (fc) | | |
| 1 conv. layer (3.1M) | 50 - 300 (c) | dependent (linear) | 100 - 4000 (fc) | | |
| 1 conv. layer (10M) | 50 - 450 (c) | dependent (linear) | 500 - 20000 (fc) | | |
| 1 conv. layer (31M) | 200 - 600 (c) | dependent (linear) | 1000 - 4100 (fc) | | |
| 2 conv. layers (1M) | 20 - 120 (c) | 20 - 120 (c) | dependent (fc) | | |
| 2 conv. layers (3.1M) | 50 - 250 (c) | 20 - 120 (c) | dependent (fc) | | |
| 2 conv. layers (10M) | 50 - 350 (c) | 20 - 120 (c) | dependent (fc) | | |
| 2 conv. layers (31M) | 50 - 800 (c) | 20 - 120 (c) | dependent (fc) | | |
| 3 conv. layers (1M) | 20 - 110 (c) | 20 - 110 (c) | 20 - 110 (c) | dependent (fc) | |
| 3 conv. layers (3.1M) | 40 - 200 (c) | 40 - 200 (c) | 40 - 200 (c) | dependent (fc) | |
| 3 conv. layers (10M) | 50 - 350 (c) | 50 - 350 (c) | 50 - 350 (c) | dependent (fc) | |
| 3 conv. layers (31M) | 50 - 650 (c) | 50 - 650 (c) | 50 - 650 (c) | dependent (fc) | |
| 4 conv. layers (1M) | 25 - 100 (c) | 25 - 100 (c) | 25 - 100 (c) | 25 - 100 (c) | dependent (fc) |
| 4 conv. layers (3.1M) | 50 - 150 (c) | 50 - 150 (c) | 50 - 200 (c) | 50 - 200 (c) | dependent (fc) |
| 4 conv. layers (10M) | 50 - 300 (c) | 50 - 300 (c) | 50 - 350 (c) | 50 - 350 (c) | dependent (fc) |
| 4 conv. layers (31M) | 50 - 500 (c) | 50 - 500 (c) | 50 - 650 (c) | 50 - 650 (c) | dependent (fc) |

### 6.2 DETAILS OF TRAINING MODELS OF VARIOUS DEPTHS ON CIFAR-10 HARD 0/1 LABELS

Models in the first four rows in Table 1 are trained similarly to those in Section 6.1, and are architecturally equivalent to the four convolutional student models shown in Table 2 with 10 million parameters. The following hyperparameters are optimized: initial learning rate $[0.0015, 0.025]$ (optimized on a log scale), momentum $[0.68, 0.97]$ (optimized on a log scale), constants $C_1, C_2 \in [0, 1]$ that control the number of filters or neurons in different layers, and up to four different dropout rates $DOc_1 \in [0.05, 0.4], DOc_2 \in [0.1, 0.6], DOc_3 \in [0.1, 0.7], DOf_1 \in [0.1, 0.7]$ for the different layers. Weight decay was set to $2 \cdot 10^{-4}$ and we used the same data augmentation settings as for the student models. We use $5 \times 5$ convolutional filters, one nonlinear hidden layer in each model and each max-pooling operation is followed by dropout with a separately optimized rate. We use $2 \times 2$ max-pooling except in the model with only one convolutional layer where we apply $3 \times 3$ pooling as this seemed to boost performance and reduces the number of parameters.

### 6.3 DETAILS OF TRAINING STUDENT MODELS OF VARIOUS DEPTHS ON ENSEMBLE LABELS

Our student models have the same architecture as models in Section 6.2. The model without convolutional layers consists of one linear layer that acts as a bottleneck followed by a hidden layer of ReLU units. The following hyperparameters are optimized: initial learning rate $[0.0013, 0.016]$ (optimized on a log scale), momentum $[0.68, 0.97]$ (optimized on a log scale), input-scale $\in [0.8, 1.25]$, global initialization scale (after initialization) $\in [0.4, 2.0]$, layer-width constants $C_1, C_2 \in [0, 1]$ that control the number of filters or neurons. The exact ranges for the number of filters and implicitly resulting number of hidden units was chosen for all twenty optimization experiments independently, as architectures, number of units and number of parameters strongly interact.

For the non-convolutional models we chose a slightly different hyper-parameterization. Given that all layers (in models with "two layers" or more) are nonlinear and fully connected we treat all of them similarly from the hyperparameter-optimizer's point of view. In order to smoothly enforce the parameter budgets without rejecting any samples from the Bayesian optimizer we instead optimize the ratios of hidden units in each layer (numbers between 0 and 1), and then re-normalize and scale them to the final number of neurons in each layer to match the target parameter budget.

Figure 2 is similar to 1 but includes preliminary results from experiments for models with 100M parameters. We are also running experiments with 300M parameters. Unfortunately, Bayesian optimization on models with 100M and 300M parameters is even more expensive than for the other points in the graph.

As expected, adding capacity to the convolutional students (top of the figure) modestly increases their accuracy. Preliminary results for the MLPs however (too preliminary to include in the graph) may not show the same increase in accuracy with increasing model size. Models with two or three hidden layers may benefit from adding capacity to each layer, but we have yet to see any benefit from adding capacity to the MLPs with four or five hidden layers.

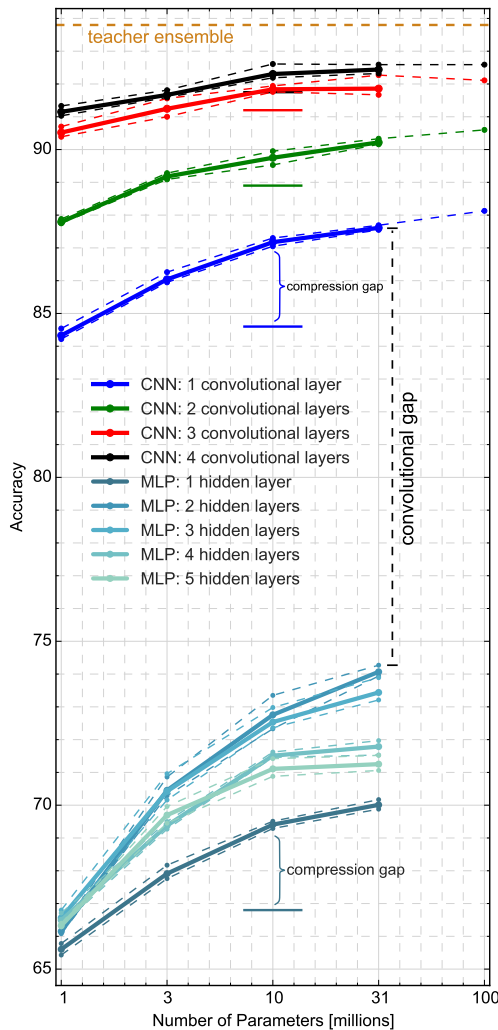

Figure 2: See figure 1.

