# Peer review of "Do Deep Convolutional Nets Really Need to be Deep and Convolutional?"

_ICLR 2017 — accepted_

[Official Review · AnonReviewer1 · rating 8 · confidence 5 · 14 Dec 2016]
**Careful study proving, to the extent possible, a fascinating point**

This paper describes a careful experimental study on the CIFAR-10 task that uses data augmentation and Bayesian hyperparameter optimization to train a large number of high-quality, deep convolutional network classification models from hard (0-1) targets.  An ensemble of the 16 best models is then used as a teacher model in the distillation framework, where student models are trained to match the averaged logits from the teacher ensemble.  Data augmentation and Bayesian hyperparameter optimization is also applied in the training of the student models.  Both non-convolutional (MLP) and convolutional student models of varying depths and parameter counts are trained.  Convolutional models with the same architecture and parameter count as some of the convolutional students are also trained using hard targets and cross-entropy loss.  The experimental results show that convolutional students with only one or two convolutional layers are unable to match the results of students having more convolutional layers under the constraint that the number of parameters in all students is kept constant.

Pros
+ This is a very thorough and well designed study that make use of the best existing tools to try to answer the question of whether or not deep convolutional models need both depth and convolution.
+ It builds nicely on the preliminary results in Ba & Caruana, 2014.

Cons
- It is difficult to prove a negative, as the authors admit.  That said, this study is as convincing as possible given current theory and practice in deep learning.

Section 2.2 should state that the logits are unnormalized log-probabilities (they don't include the log partition function).

The paper does not follow the ICLR citation style.  Quoting from the template:  "When the authors or the publication are included in the sentence, the citation should not be in parenthesis (as in “See Hinton et al. (2006) for more information.”). Otherwise, the citation should be in parenthesis (as in “Deep learning shows promise to make progress towards AI (Bengio & LeCun, 2007).”)."

There are a few minor issues with English usage and typos that should be cleaned up in the final manuscript.

necessary when training student models with more than 1 convolutional layers → necessary when training student models with more than 1 convolutional layer

remaining 10,000 images as validation set → remaining 10,000 images as the validation set

evaluate the ensemble’s predictions (logits) on these samples, and save all data → evaluated the ensemble’s predictions (logits) on these samples, and saved all data

more detail about hyperparamter optimization → more detail about hyperparameter optimization

We trained 129 deep CNN models with spearmint → We trained 129 deep CNN models with Spearmint

The best model obtained an accuracy of 92.78%, the fifth best achieved 92.67%. → The best model obtained an accuracy of 92.78%; the fifth best achieved 92.67%.

the sizes and architectures of three best models → the sizes and architectures of the three best models

clearly suggests that convolutional is critical →  clearly suggests that convolution is critical

similarly from the hyperparameter-opimizer’s point of view → similarly from the hyperparameter-optimizer’s point of view

[Official Review · AnonReviewer3 · rating 7 · confidence 3 · 16 Dec 2016]
**Experimental paper with interesting results. Well written. Solid experiments.**

Description.
This paper describes experiments testing whether deep convolutional networks can be replaced with shallow networks with the same number of parameters without loss of accuracy. The experiments are performed on he CIFAR 10 dataset where deep convolutional teacher networks are used to train shallow student networks using L2 regression on logit outputs.  The results show that similar accuracy on the same parameter budget can be only obtained when multiple layers of convolution are used. 

Strong  points.
- The experiments are carefully done with thorough selection of hyperparameters. 
- The paper shows interesting results that go partially against conclusions from the previous work in this area (Ba and Caruana 2014).
- The paper is well and clearly written.

Weak points:
- CIFAR is still somewhat toy dataset with only 10 classes. It would be interesting to see some results on a more challenging problem such as ImageNet. Would the results for a large number of classes be similar?

Originality:
- This is mainly an experimental paper, but the question it asks is interesting and worth investigation. The experimental results are solid and provide new insights.

Quality:
- The experiments are well done.

Clarity:
- The paper is well written and clear.

Significance:
- The results go against some of the conclusions from previous work, so should be published and discussed.

Overall:
Experimental paper with interesting results. Well written. Solid experiments.

[Official Review · AnonReviewer4 · rating 7 · confidence 4 · 16 Dec 2016]
**Experimental comparison of shallow, deep, and (non)-convolutional architectures with a fixed parameter budget**

This paper aims to investigate the question if shallow non-convolutional networks can be as affective as deep convolutional ones for image classification, given that both architectures use the same number of parameters. 
To this end the authors conducted a series of experiments on the CIFAR10 dataset.
They find that there is a significant performance gap between the two approaches, in favour of deep CNNs. 
The experiments are well designed and involve a distillation training approach, and the results are presented in a comprehensive manner.
They also observe (as others have before) that student models can be shallower than the teacher model from which they are trained for comparable performance.

My take on these results is that they suggest that using (deep) conv nets is more effective, since this model class encodes a form of a-prori or domain knowledge that images exhibit a certain degree of translation invariance in the way they should be processed for high-level recognition tasks. The results are therefore perhaps not quite surprising, but not completely obvious either.

An interesting point on which the authors comment only very briefly is that among the non-convolutional architectures the ones using 2 or 3 hidden layers outperform those with 1, 4 or 5 hidden layers. Do you have an interpretation / hypothesis of why this is the case? It  would be interesting to discuss the point a bit more in the paper.

It was not quite clear to me why were the experiments were limited to use  30M parameters at most. None of the experiments in Figure 1 seem to be saturated. Although the performance gap between CNN and MLP is large, I think it would be worthwhile to push the experiment further for the final version of the paper.

The authors state in the last paragraph that they expect shallow nets to be relatively worse in an ImageNet classification experiment. 
Could the authors argue why they think this to be the case? 
One could argue that the much larger training dataset size could compensate for shallow and/or non-convolutional choices of the architecture. 
Since MLPs are universal function approximators, one could understand architecture choices as expressions of certain priors over the function space, and in a large-data regimes such priors could be expected to be of lesser importance.
This issue could for example be examined on ImageNet when varying the amount of training data.
Also, the much higher resolution of ImageNet images might have a non-trivial impact on the CNN-MLP comparison as compared to the results established on the CIFAR10 dataset.

Experiments on a second data set would also help to corroborate the findings, demonstrating to what extent such findings are variable across datasets.

[Final Decision · Program Chairs · 06 Feb 2017]
**ICLR committee final decision**

The reviewers unanimously recommend accepting this paper.